# Effects of Apocynin, a NADPH Oxidase Inhibitor, in the Protection of the Heart from Ischemia/Reperfusion Injury

**DOI:** 10.3390/ph16040492

**Published:** 2023-03-27

**Authors:** Ali Mohammad, Fawzi Babiker, Maie Al-Bader

**Affiliations:** Department of Physiology, Faculty of Medicine, Kuwait University, P.O. Box 24923, Kuwait City 13110, Kuwait

**Keywords:** ischemia/reperfusion, apocynin, NOX, CD38, NAADP

## Abstract

Ischemia and perfusion (I/R) induce inflammation and oxidative stress, which play a notable role in tissue damage. The aim of this study was to investigate the role of an NADPH oxidase inhibitor (apocynin) in the protection of the heart from I/R injury. Hearts isolated from Wistar rats (n = 8 per group) were perfused with a modified Langendorff preparation. Left ventricular (LV) contractility and cardiovascular hemodynamics were evaluated by a data acquisition program, and infarct size was evaluated by 2,3,5-Triphenyl-2H-tetrazolium chloride (TTC) staining. Furthermore, the effect of apocynin on the pro-inflammatory cytokines (IL-1β, IL-6 and TNF-α) and anti-inflammatory cytokine (IL-10) was evaluated using an enzyme linked immunosorbent assay (ELISA). Hearts were subjected to 30 min of regional ischemia, produced by ligation of the left anterior descending (LAD) coronary artery, followed by 30 min of reperfusion. Hearts were infused with apocynin before ischemia, during ischemia or at reperfusion. To understand the potential pathways of apocynin protection of the heart, a nitric oxide donor (S-nitroso-N-acetylpenicillamine, SNAP), nitric oxide blocker (N (gamma)-nitro-L-arginine methyl ester, L-Name), nicotinic acid adenine dinucleotide phosphate (NAADP) inhibiter (Ned-K), cyclic adenosine diphosphate ribose (cADPR) agonist, or CD38 blocker (Thiazoloquin (az)olin (on)e compound, 78c) was infused with apocynin. Antioxidants were evaluated by measuring superoxide dismutase (SOD) and catalase (CAT) activity. Apocynin infusion before ischemia or at reperfusion protected the heart by normalizing cardiac hemodynamics and decreasing the infarct size. Apocynin treatment resulted in a significant (*p* < 0.05) decrease in pro-inflammatory cytokine levels and a significant increase (*p* < 0.05) in anti-inflammatory and antioxidant levels. Apocynin infusion protected the heart by improving LV hemodynamics and coronary vascular dynamics. This treatment decreased the infarct size and inflammatory cytokine levels and increased anti-inflammatory cytokine and antioxidant levels. This protection follows a pathway involving CD38, nitric oxide and acidic stores.

## 1. Introduction

Cardiovascular diseases (CVDs) are considered the leading cause of morbidity and death worldwide by the American Heart Association (AHA) [1]. They were reported to have led to more than 15 million deaths from 2006 to 2016 [2]. Among CVDs, coronary vascular disease (CAD) accounted for nearly seven million deaths in 2010 [3]. However, mortality rates are progressively declining due to the introduction of advanced prevention and treatment procedures [4]. Myocardial reperfusion procedure is the standard therapy for the protection of the heart against ischemia/reperfusion (I/R) injury. Nevertheless, complications such as in-hospital mortality, recurrent myocardial infarction, and left ventricular (LV) remodeling resulting in heart failure still exist [5].

Apocynin or acetovanillone is a non-toxic molecule extracted from Picrorhiza kurroa plant roots [6]. This molecule has been traditionally used to treat heart problems and diseases [7]. Several studies reported the inhibition of apocynin to nicotinamide adenine dinucleotide phosphate (NADPH)-oxidase (NOX), which reduces oxidative stress [7]. The mechanism that induces this inhibition is not fully understood. However, studies have demonstrated that apocynin disrupts the activity of NOX by preventing its assembly and ultimately decreases superoxide production and decreases reactive oxygen species (ROS) [8]. Consequently, this process induces nitric oxide (NO) bioavailability, which was reported to be protective for the heart [9]. However, the effect of apocynin on cells requires further investigation [10]. Apocynin has been used as a therapeutic drug for treating a variety of diseases, including cancer, hypertension, atherosclerosis, and I/R [11]. Treatment with apocynin at low doses significantly decreased infarct size, possibly due to a reduction in superoxide levels [12]. Very few studies have outlined apocynin as a cardioprotective molecule [12,13]; however, the mechanism whereby apocynin protects the heart from I/R injury has not been reported. 

Nicotinamide adenine dinucleotide phosphate is physiologically crucial in many cellular functions. However, it has been reported to participate in organ and tissue injury and induces generations of antioxidants and ROS [14]. NADPH was reported as an essential component in the function and production of NOX [10]. NOX’s primary function is to produce superoxide-free radicals that contribute to ROS generation [10]. The role of NOX in myocardial I/R has been extensively studied [15]. Baseline levels of NOX were reported to be essential in I/R injury [15]. Although NOX isoforms increase ROS generation, they also contribute to myocardial protection [15]. Nevertheless, increased levels of NOX exacerbate myocardial I/R injury [16]. This was reported by Krijnen et al. [17], who demonstrated a substantial elevation in NOX2 levels in myocardial infarction patients. The increase in NOX levels augments cardiomyocyte damage by inducing apoptosis and cell death [16]. These notions indicated a direct relationship between NOX inhibition and cardiomyocyte protection [18]. Qin F. et al. [19] reported that NOX inhibition causes a notable reduction in ROS and improved cardiac function. Among other drugs, apocynin was reported to inhibit NOX [10]; however, its effect and pathways of protection are not fully understood. Other molecules, such as CD38, nicotinic acid adenine dinucleotide phosphate (NAADP), and cyclic adenine diphosphoribose (cADPR) inhibitors, which may interact with apocynin, were proven to protect the heart from I/R injury [8,20]. These drugs play an essential role in myocardial salvage and decrease the infarct size; however, they require further investigation [8,20,21,22]. For this reason, CD38, NAADP, cADPR inhibitors, or agonists will be used in combination with apocynin in this study to evaluate their interaction and potential effect on the protection of the ischemic heart. 

Cluster of differentiation 38 (CD38) is a transmembrane protein with calcium-mobilizing properties present in a variety of cells [23]. Once activated, CD38 catalyzes the metabolism of cADPR and NAADP [23]. However, its role in the heart is not completely understood. Despite the importance of CD38 in many physiological functions, its role in I/R is associated with injury [24]. This is mediated by the depletion of NADPH and NO by CD38 [25]. Moreover, activated CD38 increases uncoupled eNOS, which is responsible for superoxide production [26]. These notions provide sufficient evidence for the detrimental effect of CD38 in the ischemic heart [20,24,27]. Further to this, CD38 inhibition improved myocardial contractility and salvage in a rat model [24]. 

The compound cADPR, on the other hand is a second messenger for Ca^2+^ mobilization [28]. This highlights a potential role for it in I/R injury [29]; cADPR induces Ca^2+^ overload through TRPM2 activation in the brain and kidneys [22]. This was also reported in the heart, wherein cell death by opening mPTP and the subsequent caspase cascade is induced by cADPR [22]. For instance, Eraslan et al. [30] found that cADPR inhibition by 8-bromo-cADPR protects the myocardium from I/R. Calcium mobilization by cADPR induces hypercontracture and mPTP opening, as well as cell death in I/R [31]. 

Perhaps a more potent Ca^2+^-mobilizing second messenger is the NAADP [32]. In contrast to cADPR, NAADP’s mechanism of action is ryanodine receptor channel independent [32,33]. NAADP mediates its actions by binding to two-pore channels (TPC), specifically to TPC2 found in lysosomal membranes, and exerts its Ca^2+^-mobilizing effect in such acidic organelles [33]. This pathway is physiologically essential in cardiomyocytes under normal conditions and was proven to protect the heart from I/R [34]. In contrast, the inhibition of NAADP was reported to protect the heart from I/R injury [21]. Therefore, more studies are required to clarify its role in I/R injury. 

## 2. Results

In this study, body weights (250–300 g) and heart weights (1.4 ± 0.05 g) measured at the time of sacrifice were not significantly different across the experimental groups. Heart function was evaluated by the changes in the LV hemodynamics. These include maximum developed pressure (DPmax), left ventricular end diastolic pressure (LVEDP), and LV contractility index (±dP/dt). Coronary vascular dynamics were evaluated by measuring CF and CVR. The data were obtained at the end of reperfusion and subsequently compared to that of the ischemic periods and untreated controls. Cardiac hemodynamics were drastically affected by ischemia. Myocardial ischemia worsened DPmax, increased LVEDP and decreased ±dP/dt. It also resulted in the deterioration of the coronary vascular dynamics by improving coronary flow (CF) and worsening coronary vascular resistance (CVR).

Apocynin was infused at different times throughout the experiments to investigate the role of apocynin in the protection of the heart from I/R. It was infused before ischemia, during ischemia, and at the beginning of reperfusion. Infusion of apocynin before ischemia and at the beginning of reperfusion significantly (*p* < 0.05) improved DPmax and LVEDP compared to ischemic periods and untreated controls (Figure 1A,B). A significant (*p* < 0.05) increase in CF and decrease in CVR were observed in hearts treated with apocynin before ischemia and at the beginning of reperfusion compared to ischemic periods and untreated controls (Figure 1C,D). Moreover, ±dP/dt values were significantly improved (*p* < 0.05) compared to ischemic periods and untreated controls (Figure 1E,F). However, infusion of apocynin during ischemia did not show improvement in the LV hemodynamics and coronary vascular dynamics (Figure 1A–F). 

Infarct size was measured to further validate the role of apocynin in the protection of the heart from I/R and its effects on the cardiac hemodynamics. Infusion of apocynin before ischemia and at the beginning of reperfusion significantly (*p* < 0.05) decreased the infarct size as a percentage of LV area, compared to the untreated controls. These results were not reproduced when apocynin was infused during ischemia (Figure 2). 

To understand the possible inducers of apocynin’s protection of the heart from I/R injury, the involvement of NO in this protection was evaluated. The nitric oxide donor SNAP was used in this study in combination with apocynin. However, this did not result in a synergetic addition to the protection of the heart given by apocynin, and an infusion of SNAP did not block apocynin from protecting the heart. DPmax and LVEDP were significantly improved compared to ischemic periods and untreated controls (Figure 3A,B). Similarly, infusion of SNAP in combination with apocynin significantly (*p* < 0.05) increased CF and decreased CVR when compared to ischemic periods and untreated controls (Figure 3C,D). Infusion of the nitric oxide donor (NO) S-nitroso-N-acetylpenicillamine (SNAP) also significantly (*p* < 0.05) improved ±dP/dt compared to ischemic periods and untreated controls (Figure 3E,F). To validate the effect of NO on apocynin’s protection of the heart, the NO blocker L-Name was used in combination with apocynin. This treatment negated apocynin’s protection of the heart from I/R (Figure 3A,F). 

The transmembrane protein CD38 exerts a major role in calcium signaling. In this study, 78c, the blocker of CD38, completely abrogated apocynin’s protection of the heart (Figure 3A–F). To further illustrate the role of CD38 in the protection of the heart by apocynin cADPR, a potent messenger for calcium release from the sarcoplasmic reticulum (SR) was used. The infusion of cADPR in combination with apocynin negated the protective effects of apocynin when given alone compared to ischemic periods and untreated controls (Figure 3A–F). 

The lack of protection of the heart by cADPR highlighted a possible protective role for the other downstream effector of CD3 (NAADP). Therefore, Ned-K, a potent inhibitor of NAADP, was infused in combination with apocynin to evaluate the effect of NAADP on the apocynin’s protection of the heart from I/R injury. Interestingly, infusion of the NAADP inhibitor blocked the protective effects that were obtained from infusing apocynin alone (Figure 3A–F). 

The above-reported results were confirmed by measuring the infarct area as a percentage of the LV area. Infusion of SNAP in the presence of apocynin resulted in a significant (*p* < 0.05) decrease in the infarct size compared to untreated controls (Figure 4). The decrease in infarct size caused by apocynin alone was neutralized by a nitric oxide inhibitor (N (gamma)-nitro-L-arginine methyl ester (L-Name); Thiazoloquin (az)olin (on)e compound (78c), the inhibitor of CD38; cADPR; and NAADP, when infused with apocynin compared to the respective controls (Figure 4).

To further understand apocynin’s protection of the heart, pro-inflammatory cytokine levels were evaluated. Treatment of the heart with apocynin before ischemia and at reperfusion showed a significant decrease (*p* < 0.05) in the levels of tumor necrosis factor alpha (TNF-α), interleukin 1 beta (IL-1β) and interleukin 6 (IL-6), compared to the respective untreated controls. These changes were not seen when apocynin was infused during ischemia (Figure 5A). Infusion of SNAP together with apocynin caused a significant decrease (*p* < 0.05) in TNF-α, IL-1β and IL-6 levels and a significant increase (*p* < 0.05) in the anti-inflammatory cytokine interleukin 10 (IL-10) levels compared to the respective controls. This decrease in the pro-inflammatory cytokine levels and the increase in the anti-inflammatory cytokine levels were completely negated by the infusion of L-Name, 78c, cADPR or Ned-K compared to the respective controls (Figure 5B). 

The role of the antioxidant enzymes in the protection of the heart by apocynin was investigated. The activity of superoxide dismutase (SOD) and catalase (CAT) in the hearts treated with apocynin before ischemia and at reperfusion were significantly increased (*p* < 0.05) compared to untreated controls (Figure 6 and Figure 7). In contrast, the activity of these enzymes in the hearts treated with apocynin during ischemia was not significantly different from that of the untreated controls. Additionally, treatment of the heart with a combination of apocynin and SNAP caused a significant increase (*p* < 0.05) in SOD and CAT activity compared to untreated controls (Figure 6 and Figure 7). Moreover, treatment of the heart with a combination of apocynin and SNAP caused a significant increase in the activity of SOD and CAT (*p* < 0.05). Interestingly, the infusion of L-Name, 78c, cADPR or Ned-K in combination with apocynin voids the increase in these enzymes’ activity caused by the infusion of apocynin alone (Figure 6 and Figure 7).

## 3. Discussion

In this study, we explored the potential therapeutic effects of apocynin on I/R. The study showed that apocynin infusion before ischemia or at reperfusion conferred protection to the heart from I/R injury. This treatment showed a significant improvement of the LV hemodynamics and coronary vascular dynamics and resulted in a significant decrease in the infarct size compared to respective controls. Moreover, this treatment caused a significant decrease in the levels of the pro-inflammatory cytokines TNF-α, IL-1β, and IL-6 and increased the levels of anti-inflammatory cytokine IL-10 when compared to the respective controls. Treatment with apocynin also resulted in a significant increase in the antioxidant enzyme levels. Similar results were obtained when the heart was treated with the NO donor SNAP in combination with apocynin. In contrast, infusion of L-Name, 78c, cADPR or Ned-K in the presence of apocynin repealed these protective effects. The hallmark of this study is that apocynin protected the heart via a pathway employing NO, NAADP and acidic stores (Figure 8).

The mechanism of apocynin protection to the heart is not well understood, especially with regard to its role in myocardial I/R. The lack of literature regarding the possible signaling pathways involved in apocynin’s protection of the heart necessitates logical explanations and anticipations to explain the findings of the present study. The first explanation for apocynin’s protection of the heart is possibly its ability to induce NO generation [35]. Indeed, NO is known to have protective effects in myocardial I/R by improving blood flow [36]. Moreover, NO acts as a strong scavenger for ROS and acts as a potent inhibitor of NADPH-oxidase [8]. Apocynin, on the other hand, inhibits the assembly of NOX, which is reported to be deleterious to the ischemic heart [8]. As a result, decreased levels of NOX reduces superoxide and peroxynitrite availability and reduces the impact of stressors on the heart [9]. Our findings are in line with [13,37], which reported protection of the heart from I/R by apocynin; however, these studies did not investigate the possible protective pathways, and they anticipated the apoptosis-decreasing effect of the treatment as an end effector for the protection [37]. In this study we proved that apocynin protects the heart from I/R by a pathway that involves NO, NAADP, and acidic stores. 

To further address the potential involvement of cADPR and NAADP in apocynin’s protection of the heart against I/R, this study investigated the blockade of their upstream effector CD38 [23]. CD38 utilizes NADP to synthesize NAADP, which leads to an extensive decrease in NO availability [38]. Additionally, CD38 increases uncoupled eNOS, which could increase superoxide generation [20]. In contrast, in this study, the infusion of 78c, a selective inhibitor of CD38 in the presence of apocynin, neutralized the protection of the heart by the latter. These findings suggest that CD38 is required for the protection of the heart by apocynin. To the best of our knowledge, this cross talk between apocynin and CD38 is reported for the first time in this study. The data of this study suggest that CD38 and NAADP are essential in mediating the protective effects of apocynin and may be essential for some other medicines. However, CD38 is deleterious to the ischemic heart; in the presence of apocynin, it played an inverse role and induced protection. There is no clear explanation for this effect in the literature, but we anticipated that apocynin activated a pathway that induced NAADP but not cADPR production. 

As discussed above, the role of CD38 in the protection of the heart from I/R is evident. Thus, one of its downstream effectors cADPR and NAADP, or both, could be crucial for this protection. To further investigate this and outline a logistic pathway for this protection, we infused a cADPR agonist to see if cADPR is important in this pathway. However, it is well established that Ca^2+^ overload represents a major hallmark in myocardial I/R injury [39]. Compound cADPR, which is a Ca^2+^ mobilizing agent, was proven to be a marker for Ca^2+^ overload [22]. Interestingly, the infusion of cADPR agonist abolished the protection given by apocynin. These results are in line with Xie et al. [29], who reported detrimental effects of increased cADPR levels in the heart and reported a protective effect from a cADPR inhibitor. This notion indicates that treatment with apocynin completely blocked calcium release from the SR during ischemia—a mechanism that was reported by Zhan et al. [22]. Similar results were reported by Khalaf and Babiker [34], who found that the inhibition of Ca^2+^ release from the SR is essential in the protection of the heart from I/R injury. These notions suggest an impactful effect of SR calcium release on I/R injury, and apocynin blocks these effects. Thus, future studies are required to further identify the mechanisms governing these effects and the possible molecules that are involved in them. 

The lack of a protective role from cADPR in the protection to the heart from I/R by apocynin suggests the presence of a contribution by NAADP in this protection. The blockade of CD38 negated apocynin protection to the heart, which proves its role through a downstream effector, which is NAADP. NAADP has been reported to be a potent Ca^2+^-mobilizing agent [40]; unlike cADPR, which acts on the SR, NAADP operates through a mechanism that mobilizes Ca^2+^ in the acidic-like organelles, mainly lysosomes [33]. NAADP exerts its physiological functions by binding to TPCs located on the lysosomes [33]. Additionally, NAADP has been recognized as the first trigger for Ca^2+^ release, which is then amplified by other Ca^2+^-mobilizing molecules, such as cADPR and IP3 [41]. Having these characteristics, NAADP might play a critical role in Ca^2+^ signaling in myocardial I/R injury. Thus, manipulating NAADP indicated its possible role in the protection of the heart from I/R injury. However, more studies are required to describe the mechanism and pathways for this protection. 

The infusion of Ned-K, the specific inhibitor of NAADP to the hearts in combination with apocynin completely repleaded the protection given by apocynin to the ischemic heart. This finding indicates that acidic stores’ Ca^2+^ release, unlike Ca^2+^ from the SR, is essential for the protection of the heart from I/R injury. Apocynin seems to protect the heart by a pathway using CD38, its downstream effector NAADP, and Ca^2+^ from acidic stores. The findings of this study support our previous findings and the findings of other research groups [34,42]. To the best of our knowledge, this pathway of apocynin and NAADP is reported for the first time in this study. Our data suggest that apocynin protection is mainly mediated by Ca^2+^ release from acidic stores. 

To understand apocynin’s protection of the heart, we investigated the potential presence of NO’s role in this protection. NO has been reported as a potent molecule in the protection of the heart from I/R injury. NO improves blood flow and contractile function [43] and inhibits platelet aggregation, leukocyte adhesion and the expression of cytokines [43]. However, its role in the protection of the heart by apocynin has not been previously reported. However, the infusion of NO donor, SNAP, did not add to the protection of the heart by apocynin, and it did not antagonize it either. Infusion of the specific inhibitor of NO (L-NAME), together with apocynin at reperfusion, blunted apocynin’s protection of the heart from I/R injury. These findings prove a positive role for NO in the protection of the heart by apocynin; however, the position of the NO intervention in the protective pathway requires more investigation. NO may play a major role in this protection by opening the mitochondrial ATP-dependent K^+^ channel (mitoK_ATP_) [44,45]. The findings of this study are in line with previous studies [46,47], which showed a detrimental effect to the heart by inhibiting NO. Indeed, other studies [35,48] reported a role for NO in the protection of the heart by apocynin. These notions highlight a possible role for NO in the pathways followed by apocynin to protect the heart from I/R injury.

## 4. Materials and Methods

### 4.1. Experimental Procedure

Adult male Wister rats weighing between 250 and 300 g were used in this study. A total of 72 rats were divided into 9 groups for this study. These animals were obtained from the Animal House, Health Science Center, Kuwait University. The study was approved by the Health Sciences Center, Kuwait University Animal Ethics Committee. All procedures were conducted according to the National Institutes of Health guidelines for the care and use of laboratory animals (NIH Publication No. 85–23, Revised 1985). All animals were maintained under controlled temperature (21–24 °C), 12 h light/dark cycle (light 7 a.m.–7 p.m.) and 50% humidity (50%). They were housed in plastic cages (2 rats/cage) and allowed tap water and food unrestrictedly. All rats were anesthetized using a mixture of Ketamine (50 mg/kg) and Xylazine (5 mg/kg), which were administered intramuscularly. To protect against thrombi formation, heparin (1000 U/kg body weight) was infused through the femoral vein. The heart was excised as described previously by Khalaf and Babiker [34]. Briefly, the thoracic cavity was opened; the diaphragm was then carefully cut, and subsequently, the heart was excised and immediately immersed in cold (4 °C) Krebs-Hensleit (KH) solution. 

Immediately after the isolation process, the heart was cannulated to a modified Langendorff system, as described previously [49,50]. To fix the heart to the modified Langendorff system, a cannula was inserted in the aorta and was stabilized by a small clip. Thereafter, a ligature was secured around the aorta, and the clip was removed. The perfusion solution was instantly infused to restore normal cardiac contractility and heart rate. The heart was perfused with freshly prepared KH buffer (mM: NaCl 117.86, KCl 5.59, CaCl_2_·H_2_O 2.4, NaHCO_3_ 20, KH_2_PO_4_ 1.19, MgCl_2_·6H_2_O 1.2, Glucose 12.11). The buffer was supplied with a mixture of oxygen (95%) and carbon dioxide (5%); pH was 7.35–7.45 at a temperature of 37 ± 0.5 °C. 

The perfusion pressure (PP) was measured by a pressure transducer (P23 DB) downstream from a flow probe located on a branch of the aortic cannula. This pressure was maintained throughout the experimental procedure in all protocols at 50 mmHg. To maintain a constant PP, a perfusion assembly (Module PPCM type 671, Hugo Sachs Electroniks, Germany) was used, which provided a precise adjustment of the PP between 5 and 150 mmHg, with accuracy of ±1 mmHg. The perfusion buffer temperature was kept constant at 37 °C by circulating temperature-controlled water using a water bath (RMS Lauda, Dr. R. Wobser GmbH & Co., March, Germany) and a Techno Circulator (Cole-Parmer Instrument Company, Chicago, IL, USA). Moreover, myocardial temperature was measured using a needle thermistor probe (Thermlert TH-5, Physitemp, San Diego, CA, USA) inserted in the apex of the heart in all experiments. The hearts were instrumented with pacing electrodes on the right atrium (RA) appendages to maintain proper heart rate at the physiological heart rate of the rat. 

### 4.2. Study Protocol

All hearts were allowed to stabilize for 20 min after cannulation. This was followed by a 30 min coronary artery occlusion (CAO), which was accomplished by ligating the left anterior descending coronary artery (LAD). A snare was used to encircle the LAD at 0.5 cm below the atrioventricular groove. Additionally, a small rigid plastic tube was placed between the heart and the snare to maintain complete occlusion of the coronary artery. After ischemia, the snare was cut, and the small rigid plastic tube was removed to allow for a 30 min reperfusion, as described previously [51,52]. The animals were randomly subdivided into 9 groups (n = 8/group). The control hearts were subjected to I/R without any additional treatment. Both pre-treatment and post-treatment protocols were used in this study. In the pre-treatment protocols, apocynin (1 mM) infusion started during the stabilization period and continued for 15 min before ischemia or during ischemia and continued for 15 min. In the post-treatments, apocynin (1 mM) was infused 5 min before reperfusion and continued 10 min during reperfusion [13]. Compound 78c, the inhibitor of CD38 (10 µM), [24]; NAADP inhibitor (Ned-K) (10 µM), [21]; cADPR agonist (10 µM), nitric oxide donor SNAP (10 µM); or nitric oxide inhibitor L-Name (10 µM) [53] were infused to the heart at reperfusion in presence of apocynin (Figure 9).

### 4.3. Cardiac Hemodynamics Assessment

To evaluate heart function, LV hemodynamics and coronary-vascular dynamics were evaluated during stabilization, ischemia, and reperfusion periods. The LV dynamics were determined throughout the experiment by assessing the LVEDP, DPmax, and ±dP/dt as described previously [54]. A latex balloon was lodged in the LV through the mitral valve to measure LV pressure changes throughout the experiment. The balloon was inflated using a microsyringe that pumps buffer and was attached to a pressure transducer and a DC bridge amplifier (DC-BA) with a pressure module (DC-BA type 660, Hugo-Sachs Electronik, Germany) and connected to a personal computer for the monitoring of the hemodynamics. DPmax was measured by a Max-Min module (MMM type 668, Hugo-Sachs Electronick, Germany) that converts the DC-BA output to DPmax, by subtracting LVEDP from the LVESP. 

The evaluation of the coronary vascular dynamics was done as previously described [55]. CF was measured by an electromagnetic flow probe attached to the inflow of the aortic cannula. The probe was also connected to a flow meter that transmits signals to a personal computer. The coronary flow in ml/min was constantly monitored by manual collection of coronary effluent. The CVR was computed every 10 s by an online data acquisition program (Isoheart software V 1.524-S, Hugo-Sachs Electronik, Germany). 

### 4.4. Sample Storage

Coronary effluent was obtained from hearts in all experiments. Subsequently, samples were frozen instantly in liquid nitrogen. Similarly, all hearts were immediately frozen in liquid nitrogen at the end of the experiments. All effluent samples and hearts were stored at −80 °C for further biochemical analysis. 

### 4.5. Evaluation of Cardiac Injury by Measurements of Infarct Size and Cardiac Enzyme Levels

The LV area and the infarct area were evaluated by ImageJ (Image J, Wayne Rasb and National Institute of Health, Bethesda, MD, USA). After reperfusion, hearts were stored overnight at −20 °C and were sliced into 4–6 sections from apex to base the next day. The slices were then incubated in 1% triphenyl tetrazolium chloride (TTC) solution in isotonic (pH 7.4) phosphate buffer and then fixed overnight in 4% formaldehyde. TTC was used to distinguish metabolically viable and dead tissues. TTC reacts with NADH in viable tissues that contain various dehydrogenases, causing them to stain deep red. On the other hand, non-stained necrotic tissues possess an infarcted pale white color. The red and pale white areas of every slice were manually indicated on the image. Finally, the infarcted tissue area as a percentage of LV area was measured for all hearts. 

### 4.6. Protein Extraction

The LV tissue was homogenized in ice-cold MOPS lysis buffer (20 mM; KCl, 150 mM; Mg acetate, 4.5 mM; Triton X, 1%) containing protease inhibitor cocktail (Roche Applied Science, Mannheim, Germany), and the homogenate was centrifuged at 14,000× *g* for 15 min. The supernatant was collected, the protein content was measured with a Bio-Rad protein assay (Bio-Rad, Hercules, CA, USA), and the samples were allocated and stored for further biochemical analysis.

### 4.7. Enzyme-Linked Immunosorbent Assay (ELISA)

Left ventricular tissue was homogenized in ice-cold lysis buffer, and the homogenate was centrifuged at 4000× *g* for 10 min. The supernatant was collected, and protein content was measured. Dilution was done as required by the manufacturer’s guidelines. The protein levels of TNF-α (cat# MBS175904), IL-1β (cat# MBS825017), IL-6 (cat# MBS355410), and IL-10 (cat# MBS355232) were evaluated by ELISA. The detection procedures for each of the proteins were according to the manufacturer guidelines in the respective commercial kits. All kits were obtained from BioSource International, Camarillo, CA, USA.

### 4.8. Antioxidant Evaluation

SOD and CAT activities were measured by ultraviolet visible (UV-Vis) spectrophotometer. Measurement of SOD (cat# EC.1.15.1.1, Sigma, St Louis, MI, USA) is based on the principle of reduction of nitroblue tetrazolium (NBT) compound found in the reaction medium when superoxide radical, formed by the xanthine–xanthine oxidase system, cannot be removed by SOD enzyme; the results were expressed in terms of U/mg. One unit of SOD was expressed as the substance amount that caused 50% inhibition in the NBT reduction rate [56]. 

CAT (cat# EC 1.11.1.6, Sigma, St Louis, MI, USA) activity was measured as described previously [56]. The method is based on the direct measurement of the decrease in absorbance at 240 nm (ε240 nm = 43.6 M^−1^ cm^−1^) due to H_2_O_2_ consumption by CAT. To optimize the assay conditions, relatively low H_2_O_2_ concentration was used. Assays were carried out in 50 mM K phosphate buffer, pH 7.0, with 10 mM H_2_O_2_ at 25 °C. CAT-specific activity was expressed as units of CAT by mg of protein of enzymatic preparation. One CAT unit was defined as the mmol of H_2_O_2_ consumed per min. 

### 4.9. Data Analysis

All data are represented as mean ± SEM. Two-way analysis of variance (ANOVA) for repeated measures within each group and between the groups was performed on absolute values, even when presented as a percentage of baseline. Only when analysis showed a significant difference was post-hoc analysis with Tukey’s correction used for further comparison. Student’s *t*-Test was used to assess the significance in biochemical experiments, infarct size, pre-inflammatory cytokines and anti-inflammatory cytokines (Microsoft Excel). A *p* < 0.05 was considered statistically significant.

## 5. Conclusions

Treatment with apocynin before ischemia or at reperfusion protected the heart from I/R injury. This protection reduced infarct size and proinflammatory cytokines and increased anti-inflammatory cytokine and antioxidant levels. This protection seems to follow a pathway involving CD38, NO, NAADP and the acidic stores. We recommend follow up studies to understand the role of acidic stores’ calcium in the protection of the heart from I/R injury.

## Figures and Tables

**Figure 1 pharmaceuticals-16-00492-f001:**
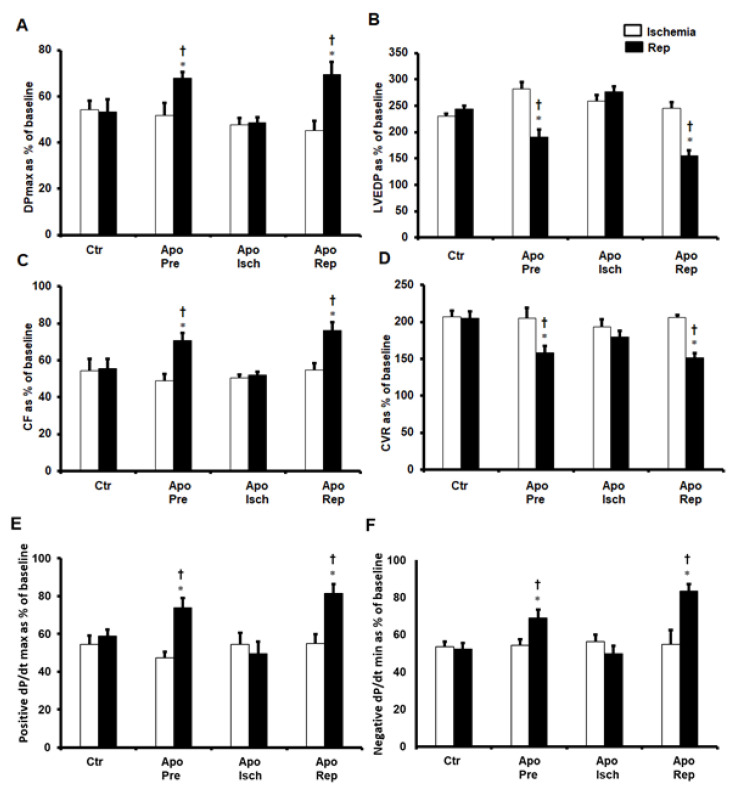
Post-ischemic recovery in cardiac hemodynamics. Cardiac hemodynamics when apocynin was infused before ischemia, during ischemia, or at the beginning of reperfusion, presented as percentage of baseline (n = 8 per group). Data were obtained at the end of reperfusion and presented as means ± SEM. DPmax (**A**), maximum developed pressure; LVEDP (**B**), left ventricular end-diastolic pressure; CF (**C**), coronary flow; CVR (**D**), coronary vascular resistance; positive and negative dP/dt (**E**,**F** respectively), contractility index; Ctr, control; Apo, apocynin; Pre, before ischemia; Isch, during ischemia; Rep, at reperfusion. * *p* < 0.05 compared to controls. † *p* < 0.05 compared to ischemia.

**Figure 2 pharmaceuticals-16-00492-f002:**
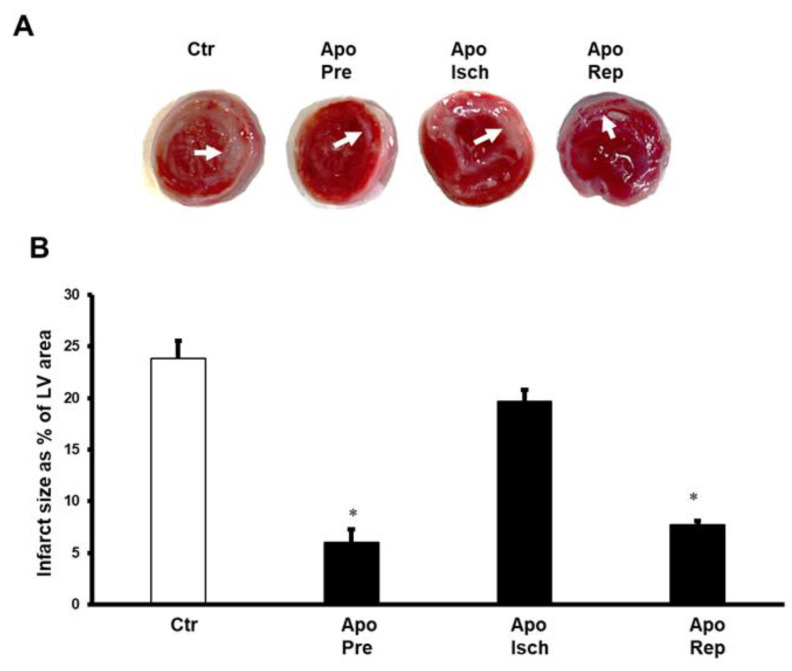
Infarct size presented as percentage of LV area. Evaluation of infarct size when apocynin was infused before ischemia, during ischemia, and at the beginning of reperfusion (n = 4 per group). Data are shown as means ± SEM. Ctr, control; Apo, apocynin; Pre, before ischemia; Isch, during ischemia; Rep, at reperfusion. (**A**) shows the TTC staining for the evaluation of the infarct size, and (**B**) shows the infarct size as percentage of LV area. * *p* < 0.05 compared to controls.

**Figure 3 pharmaceuticals-16-00492-f003:**
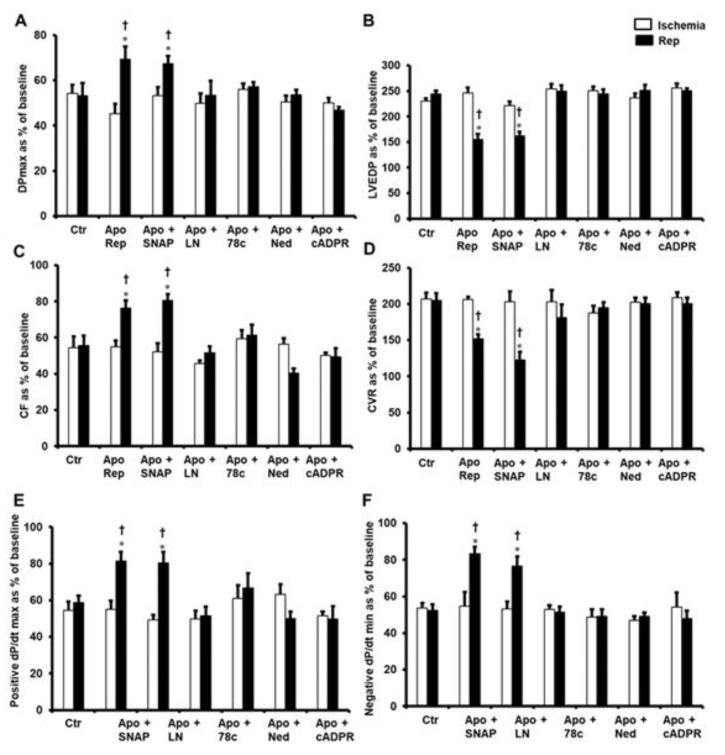
Post-ischemic recovery in cardiac hemodynamics after infusion of apocynin together with other inhibitors or agonists at reperfusion. Cardiac hemodynamics after the infusion of a combination of apocynin with SNAP, L-NAME, 78c, Ned, or cADPR agonist, presented as percentage of baseline (n = 8 per group). Data were obtained at the end of reperfusion and presented as means ± SEM. DPmax (**A**), maximum developed pressure; LVEDP (**B**), left ventricular end-diastolic pressure; CF (**C**), coronary flow; CVR (**D**), coronary vascular resistance; positive and negative dP/dt (**E**,**F** respectively), contractility index; Ctr, controls; Apo, apocynin; SNAP, nitric oxide donor; LN, L-NAME; 78c, CD38 blocker; Ned, NAADP inhibitor; cADPR, cADPR agonist. * *p* < 0.05 compared to controls. † *p* < 0.05 compared to ischemia.

**Figure 4 pharmaceuticals-16-00492-f004:**
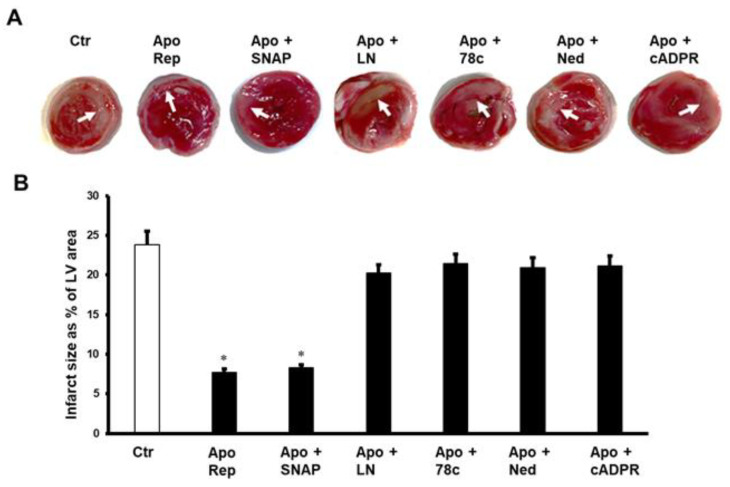
Infarct size as percentage of LV area after infusion of apocynin together with other inhibitors or agonists at reperfusion. Evaluation of infarct size in hearts treated with a combination of apocynin with SNAP, L-NAME, 78cr, Ned, or cADPR agonist (n = 4 per group). Data are shown as means ± SE. Ctr, controls; Apo, apocynin; SNAP, nitric oxide donor, LN, L-NAME; 78c, CD38 blocker; Ned, NAADP inhibitor; cADPR, cADPR agonist. (**A**) shows the TTC staining for the evaluation of the infarct size, and (**B**) shows the infarct size as percentage of LV area. * *p* < 0.05 compared to controls.

**Figure 5 pharmaceuticals-16-00492-f005:**
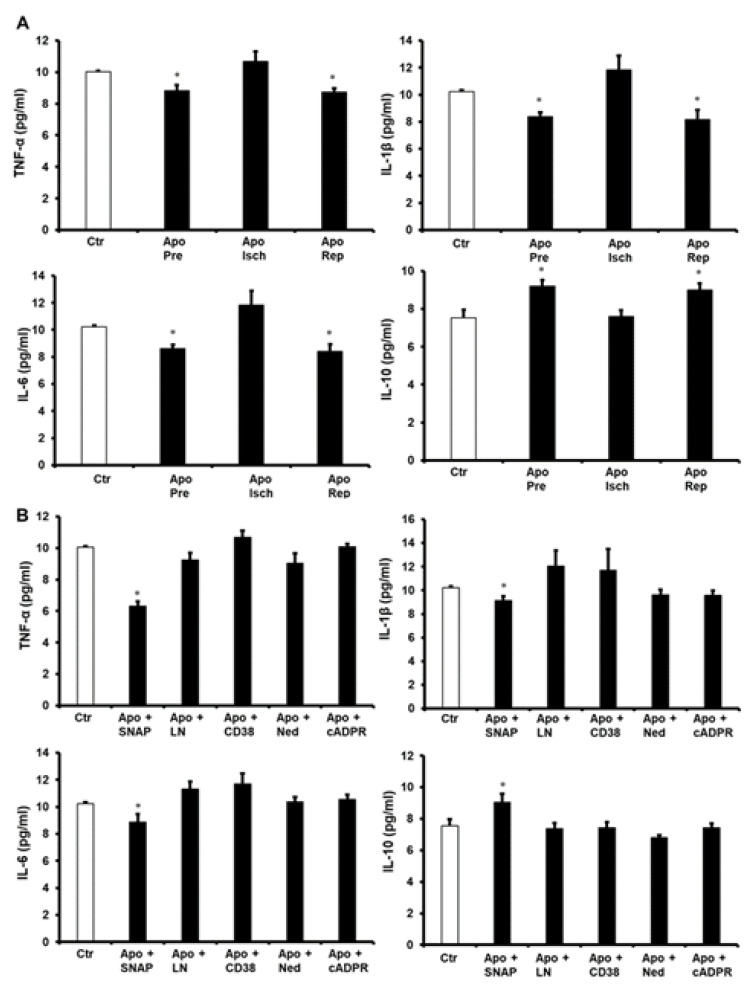
The levels of pro-inflammatory and anti-inflammatory cytokines in heart homogenates in all experiments. Quantification of TNF-α, IL-1β, IL-6 and IL-10 in hearts treated with apocynin (**A**) or a combination of apocynin and SNAP, L-NAME, 78c, Ned, or cADPR agonist (**B**) (n = 4 per group). Data are shown as means ± SEM. Ctr, control; Apo, apocynin; Pre, before ischemia, Isch, during ischemia; Rep, at reperfusion; SNAP, nitric oxide donor, LN, L-NAME; 78c, CD38 blocker; Ned, NAADP inhibitor; cADPR, cADPR agonist. * *p* < 0.05 compared to controls.

**Figure 6 pharmaceuticals-16-00492-f006:**
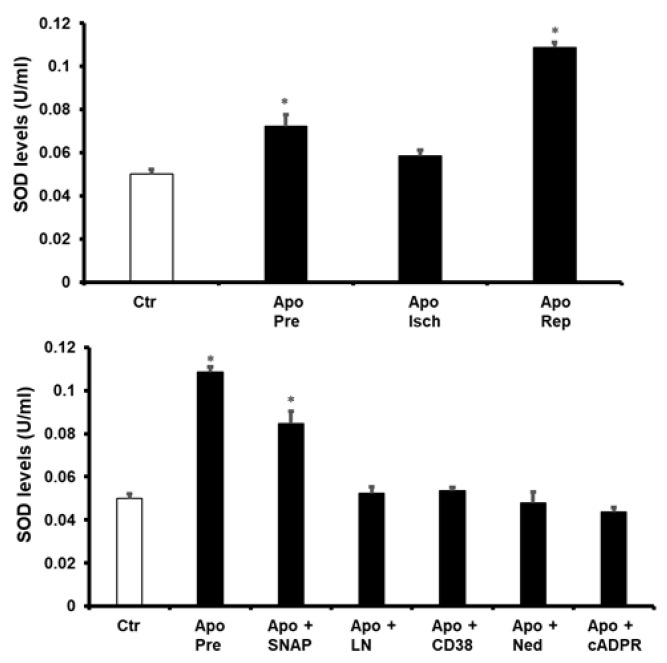
The levels of the antioxidant enzyme SOD in heart homogenates in all experiments. Quantification of SOD in hearts treated with apocynin or a combination of apocynin and SNAP, L-NAME, 78c, Ned, or cADPR agonist (n = 4 per group). Data are shown as means ± SEM. Ctr, control; Apo, apocynin; Pre, apocynin before ischemia; Isch, apocynin during ischemia; Rep, apocynin at reperfusion; SNAP, nitric oxide donor, LN, L-NAME; 78c, CD38 blocker; Ned, NAADP inhibitor; cADPR agonist. * *p* < 0.05 compared to controls.

**Figure 7 pharmaceuticals-16-00492-f007:**
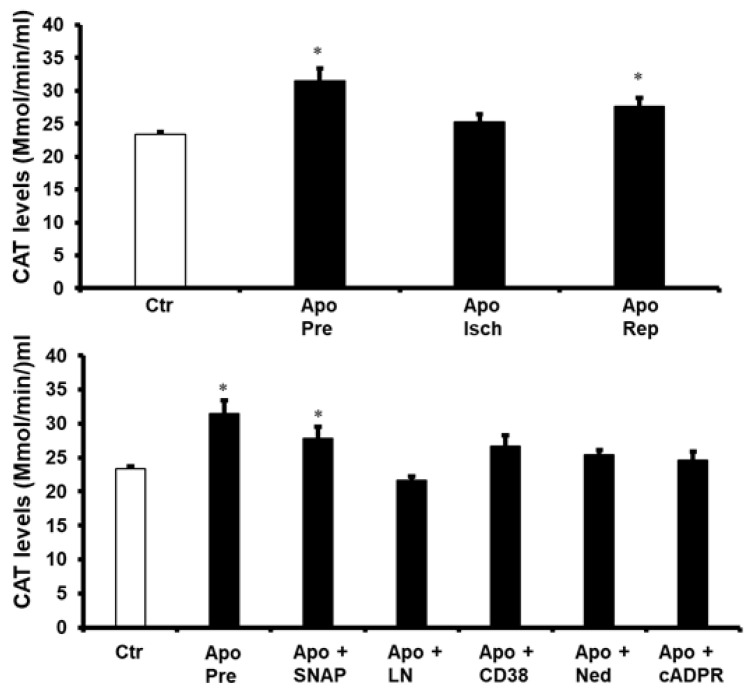
The levels of the antioxidant CAT in heart homogenates in all experiments. Quantification of SOD in hearts treated with apocynin or a combination of apocynin and SNAP, L-NAME, 78c, Ned, or cADPR agonist (n = 4 per group). Data are shown as means ± SEM. Ctr, control; Apo, apocynin; Pre, apocynin before ischemia; Isch, apocynin during ischemia; Rep, apocynin at reperfusion; SNAP, nitric oxide donor, LN, L-NAME; 78c, CD38 blocker; Ned, NAADP inhibitor; cADPR, cADPR agonist. * *p* < 0.05 compared to controls.

**Figure 8 pharmaceuticals-16-00492-f008:**
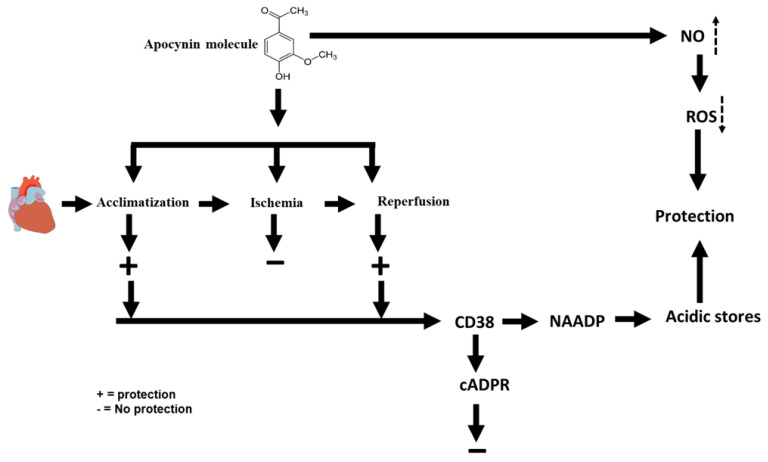
Schematic representation showing the protective treatments of apocynin and the potential pathways for this protection. NO, nitric oxide, ROS, reactive oxygen species; NAADP, nicotinic acid adenine dinucleotide phosphate; cADPR, cyclic adenine diphosphoribose: CD38, cluster of differentiation 38. (Upward pointing dotted arrow indicate increase in NO and downward pointing dotted arrow indicates decrease in ROS).

**Figure 9 pharmaceuticals-16-00492-f009:**
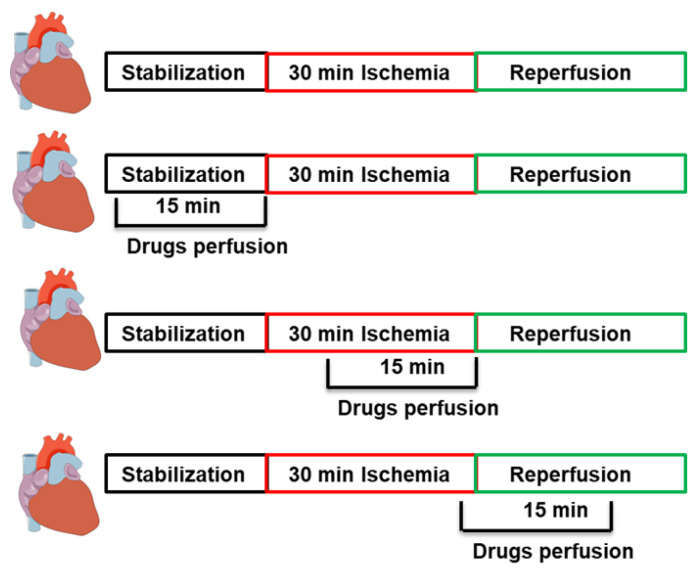
Schematic illustration showing the treatment protocols. Infusion of drugs before ischemia, during ischemia, or at reperfusion for 15 min each.

## Data Availability

Data are contained within the article.

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
