# Peer review of "Effects of Apocynin, a NADPH Oxidase Inhibitor, in the Protection of the Heart from Ischemia/Reperfusion Injury"

_pharmaceuticals, 2023, doi:10.3390/ph16040492_

Round 1

Reviewer 1 Report

The therapeutic potential of apocynin in I/R was investigated by the authors. Apocynin significantly reduced the extent of the infarct relative to the corresponding controls and demonstrated a substantial improvement in the LV hemodynamics and coronary vascular dynamics. Pro-inflammatory cytokine levels were markedly lowered, and anti-inflammatory cytokine levels were elevated as a result of apocynin administration. The authors also demonstrated that apocynin causes an increase in antioxidant enzymes.

The paper is well-written and presents novel results. For be suitable for publication, it suggests modifying some details.

I suggest fixing all the graph boxes. They are very separated between them, resulting in wasted space and lower resolution.

Figure 1: Please increase the font size of the Y axis to make it readable.

Figure 2: Please isolate in two figures. Figure A, to show only the hearts/LV in a box and identify with arrows the infarct section. The image is badly cropped. There is little neatness in cutting the images. Make homogeneous cuts or show the figure without modifying it. Figure B, the graph.

Figure 3: Please increase the font size of the Y axis to make it readable.

Figure 4: : Please isolate in two figures. Figure A, to show only the hearts/LV in a box, and identify with arrows the infarct section. The image is badly cropped. There is little neatness in cutting the images. Make homogeneous cuts or show the figure without modifying it. Figure B, the graph.

Figure 5: Please increase the font size of the Y axis to make it readable.

Figure 6: There is no name on the Y axis, only concentration. please include the name of what was measured.

Figure 7: There is no name on the Y axis, only concentration. please include the name of what was measured.

Figure 8: Please improve the quality of the diagram. I suggest using software to better understand the proposed mechanisms.

Figure 9: Please improve the quality of the diagram.

Author Response

Response to reviewer #1

The therapeutic potential of apocynin in I/R was investigated by the authors. Apocynin significantly reduced the extent of the infarct relative to the corresponding controls and demonstrated a substantial improvement in the LV hemodynamics and coronary vascular dynamics. Pro-inflammatory cytokine levels were markedly lowered, and anti-inflammatory cytokine levels were elevated as a result of apocynin administration. The authors also demonstrated that apocynin causes an increase in antioxidant enzymes.

The paper is well-written and presents novel results. For be suitable for publication, it suggests modifying some details.

 We would like to thank the reviewer for the constructive critiques and suggestions. We believe that the revised manuscript has been greatly improved after the changes we made according to the reviewer’s suggestions. All changes made in the manuscript are given in trackable changes format. We thank the reviewer for acknowledging the novelty and the importance of our study. Below are our point-by-point responses to the reviewers’ suggestions:

I suggest fixing all the graph boxes. They are very separated between them, resulting in wasted space and lower resolution.

We would like to thank the reviewer for his constructive suggestion and apologize for the way we presented our results. We improved the graphs and decreases the separation between them to the possible maximum as recommended by the first and the second reviewers.  

Figure 1: Please increase the font size of the Y axis to make it readable.

The font size of the Y axis was increased as suggested by the reviewer.

Figure 2: Please isolate in two figures. Figure A, to show only the hearts/LV in a box and identify with arrows the infarct section. The image is badly cropped. There is little neatness in cutting the images. Make homogeneous cuts or show the figure without modifying it. Figure B, the graph.

We thank the reviewer for these invaluable comments which we believe will improve the presentation of our results. We apologize for the quality of the cropped images which was imposed by presence of multiple pictures in one image during the evaluation and measurements. Now we cropped them in a homogenous way as recommended by the reviewer.

Figure 3: Please increase the font size of the Y axis to make it readable.

The font size of the Y axis was increased as suggested by the reviewer.

Figure 4: Please isolate in two figures. Figure A, to show only the hearts/LV in a box, and identify with arrows the infarct section. The image is badly cropped. There is little neatness in cutting the images. Make homogeneous cuts or show the figure without modifying it. Figure B, the graph.

We thank the reviewer for these valuable comments which we believe will improve the presentation of our results. We apologize for the quality of the cropped images which was imposed by presence of multiple pictures on one image. Now we cropped them in a homogenous way as recommended by the reviewer.

Figure 5: Please increase the font size of the Y axis to make it readable.

The font size of the Y axis was increased as requested by the reviewer.

Figure 6: There is no name on the Y axis, only concentration. please include the name of what was measured.

We appreciate the reviewer’s important comments, which pinpointed a serious flaw which could have made unnecessary confusion to the reader. The name was added to Y axes of the figure as suggested by the reviewer.

Figure 7: There is no name on the Y axis, only concentration. please include the name of what was measured.

The name was added to Y axes of the figure as suggested by the reviewer.

Figure 8: Please improve the quality of the diagram. I suggest using software to better understand the proposed mechanisms.

 The quality of the diagram was improved as suggested by the reviewer.

Figure 9: Please improve the quality of the diagram.

The quality of the diagram was improved as suggested by the reviewer.

Reviewer 2 Report

Relevance

The topic of research is very relevant - the search for new molecular targets for protecting the myocardium from reperfusion damage. Reperfusion injury inevitably occurs during coronary artery recanalization in patients with myocardial infarction, during cardiosurgical operations with cardiopulmonary bypass. Subsequently, myocardial injury leads to the development of myocardial fibrosis, heart failure, and poor prognosis for the patient.

Novelty

Thus, on the one hand, it is widely known that ROS play a significant role in damage to myocardial cells during reperfusion; that NOX is one of the important sources of superoxide anion, and inhibition of this enzyme leads to cardioprotection. However, this article discloses for the first time important aspects of the mechanism of NOX inhibition, such as the activation of NOS, elevate of antioxidant enzymes activity, inhibition of CD38, etc. Such data have not been previously published.

The design of the article corresponds to the purpose of the study. The conclusions made by the authors are adequate to the results obtained. There are no fundamental remarks.

Minor remarks:

Line 52-53: is this sentence about cerebral ischemic stroke or myocardial infarction? References [12, doi: 10.1016/j.neuroscience.2008.03.090] is about protective effect of apocynin at the cerebral stroke.

Line 54: reference 14 not content data about cardioprotection of apocynin, only about levels of ADMA, MPO, iNOS and TLR4, that is not equality.

Line 56-57: Not understand sentence: “Nicotinamide adenine dinucleotide phosphate is physiologically crucial in many cel-56 lular functions; once synthetized, it induces generations of antioxidants and ROS”

Line 64: It should be indicate, that Krijnen et al. shown Nox2 expression evaluation.

Line 83: [26] not relevant citing. These data related to PASMCs, should give a relevant siting to cardiomyocytes

Line 112: “improved LVEDP” – not appropriate phrase, will be better “increased LVDEP”

Fig 3,4, 6 (bottom diagram), 7(bottom diagram) should indicate bar “APO” and significance of differences to “APO” group.

Line 424-427 It should indicate, pre-treatment protocols apocynin or post-treatment protocol was used. 

Fig. 8 and discussion. Should direct on the figure 8, that NO decreases ROS. And, I think, we can discuss other mechanisms of NO action in cardiomyocytes - protein nitrosylation, PCG activation, which leads to the opening of mitochondrial K-ATP channels.

Author Response

Response to reviewer #2

The topic of research is very relevant - the search for new molecular targets for protecting the myocardium from reperfusion damage. Reperfusion injury inevitably occurs during coronary artery recanalization in patients with myocardial infarction, during cardiosurgical operations with cardiopulmonary bypass. Subsequently, myocardial injury leads to the development of myocardial fibrosis, heart failure, and poor prognosis for the patient.

Novelty

Thus, on the one hand, it is widely known that ROS play a significant role in damage to myocardial cells during reperfusion; that NOX is one of the important sources of superoxide anion, and inhibition of this enzyme leads to cardioprotection. However, this article discloses for the first time important aspects of the mechanism of NOX inhibition, such as the activation of NOS, elevate of antioxidant enzymes activity, inhibition of CD38, etc. Such data have not been previously published.

We would like to thank the reviewer for his/her useful comments, which helped us to improve the quality of the manuscript. We also apologize to the reviewer for the mismatch of some of our references to the cited literature. All the reviewer’s comments have been addressed, and corresponding changes have been made directly to the manuscript where appropriate. All changes made in the manuscript are highlighted as trackable changes. We thank the reviewer for acknowledging the novelty of our study.

The design of the article corresponds to the purpose of the study. The conclusions made by the authors are adequate to the results obtained. There are no fundamental remarks.

Thanks to the reviewer for the encouraging comments.

Minor remarks:

Line 52-53: is this sentence about cerebral ischemic stroke or myocardial infarction? References [12, doi: 10.1016/j.neuroscience.2008.03.090] is about protective effect of apocynin at the cerebral stroke.

Thank to the reviewer for this invaluable comment. We used this reference specifically to show the role of the potential protection of apocynin in other organs which could be applicable also to the heart. Now we realize that having a reference from the literature of the ischemic heart disease is more feasible. Reference 12, 14 and 26 were changed as requested by the reviewer.

Line 54: reference 14 not content data about cardioprotection of apocynin, only about levels of ADMA, MPO, iNOS and TLR4, that is not equality.

The reference was replaced by a suitable one as suggested by the reviewer as mentioned in our response to the reviewer.

Line 56-57: Not understand sentence: “Nicotinamide adenine dinucleotide phosphate is physiologically crucial in many cel-56 lular functions; once synthetized, it induces generations of antioxidants and ROS”

We apologize to the reviewer for the confusion caused by the construction of the sentence. The sentence was now corrected to read “Nicotinamide adenine dinucleotide phosphate is physiologically crucial in many cellular functions. However, it was reported to participate in organ and tissue injury and induces generations of antioxidants and ROS.

Line 64: It should be indicate, that Krijnen et al. shown Nox2 expression evaluation.

We apologize to the reviewer for the confusing typo. The word was corrected as suggested by the reviewer.

Line 83: [26] not relevant citing. These data related to PASMCs, should give a relevant siting to cardiomyocytes

However, here we were referring to a general role for CD38 in in ischemia reperfusion and not exclusively in the heart per se. We now understand the opinion of the reviewer and his/her preference for a citation from the heart literature. Therefore, the reference was replaced by one with the same information from the literature of the heart research.

Line 112: “improved LVEDP” – not appropriate phrase, will be better “increased LVDEP”

The word improved was replaced by increased as recommended by the reviewer.

Fig 3,4, 6 (bottom diagram), 7(bottom diagram) should indicate bar “APO” and significance of differences to “APO” group.

Thanks to the reviewer for the constructive comments. The evaluation and statistical analysis were done for all the groups together, however, APO bar was not included in the bottom diagrams mentioned above by the reviewer was because we wanted to present the apocynin data separate from that of the agonists and antagonists shown in the above-mentioned bottom diagrams. We do not contradict the reviewer if he/she sees a better and easy understanding when APO bar is included. Therefore, we added APO bar with statistical significance sign as recommended by the reviewer.

Line 424-427 It should indicate, pre-treatment protocols apocynin or post-treatment protocol was used. 

We apologize to the reviewer for the lack of clarity in the statement we use. This part was now corrected as requested by the reviewer.

Fig. 8 and discussion. Should direct on the figure 8, that NO decreases ROS. And, I think, we can discuss other mechanisms of NO action in cardiomyocytes - protein nitrosylation, PCG activation, which leads to the opening of mitochondrial K-ATP channels.

We thank the reviewer for this comment. We believe that our study described a novel pathway for the protection of apocynin to the heart from ischemia reperfusion injury. The role of every molecule involved in this pathway deserves to be discussed. However, the other mechanisms of nitric oxide action are beyond the scope of our study. We therefore indicated the effect of nitric oxide on ROS clearly in figure 8 and added statement in the discussion anticipating a possible role for cardiomyocytes-protein nitrosylation PCG activation in the protection reported in our study.
